# A Mobile Health Application Using Geolocation for Behavioral Activity Tracking

**DOI:** 10.3390/s23187917

**Published:** 2023-09-15

**Authors:** Mohamed Emish, Zeyad Kelani, Maryam Hassani, Sean D. Young

**Affiliations:** 1Department of Informatics, University of California, Irvine, CA 92697-3100, USA; elkelanz@uci.edu (Z.K.); mhassan2@uci.edu (M.H.); syoung5@hs.uci.edu (S.D.Y.); 2Department of Emergency Medicine, University of California, Irvine, CA 92697-3100, USA

**Keywords:** software, mHealth, geospatial data, assisted global positioning system, mobility analysis, location-based health services, blockchain, data integration

## Abstract

The increasing popularity of mHealth presents an opportunity for collecting rich datasets using mobile phone applications (apps). Our health-monitoring mobile application uses motion detection to track an individual’s physical activity and location. The data collected are used to improve health outcomes, such as reducing the risk of chronic diseases and promoting healthier lifestyles through analyzing physical activity patterns. Using smartphone motion detection sensors and GPS receivers, we implemented an energy-efficient tracking algorithm that captures user locations whenever they are in motion. To ensure security and efficiency in data collection and storage, encryption algorithms are used with serverless and scalable cloud storage design. The database schema is designed around Mobile Advertising ID (MAID) as a unique identifier for each device, allowing for accurate tracking and high data quality. Our application uses Google’s Activity Recognition Application Programming Interface (API) on Android OS or geofencing and motion sensors on iOS to track most smartphones available. In addition, our app leverages blockchain and traditional payments to streamline the compensations and has an intuitive user interface to encourage participation in research. The mobile tracking app was tested for 20 days on an iPhone 14 Pro Max, finding that it accurately captured location during movement and promptly resumed tracking after inactivity periods, while consuming a low percentage of battery life while running in the background.

## 1. Introduction

Geolocation tracking systems can unlock valuable insights into spatial health informatics by providing researchers and healthcare professionals with a rich source of health-related data [1]. The geographic distributions of geolocation data can be used to design more targeted and effective public health interventions, allocate resources to impoverished areas, and develop strategies for improving health outcomes for specific communities or populations [2]. Additionally, geolocation tracking systems can facilitate real-time monitoring of disease outbreaks and other health emergencies, allowing for more timely and effective response efforts [3,4]. On an individual level, geolocation data can be used to develop personalized health interventions for at-risk patients and provide well-being recommender systems with user context information [5,6]. A previous study has demonstrated that individual-level cell phone mobility data are an effective tool in assessing the effectiveness of online health interventions by monitoring changes in adherence to stay-at-home orders after exposure to targeted public health advertisements [7].

Smartphones have become an essential part of our daily lives, offering communication and a plethora of sensors that can be used to collect accurate data. With built-in GPS, accelerometers, gyroscopes, and other sensors, smartphones can provide detailed information about a user’s location, movement, and environment [8]. These data can be used in healthcare to gather previously tricky or impossible insights. For example, smartphone sensors can track physical activity, monitor sleep patterns, measure air quality, and detect noise pollution [9]. Moreover, artificial intelligence and machine learning advances have made analyzing and interpreting these data easier and more efficient [10]. For example, a study utilized machine learning and geolocation data from smartphones to investigate the impact of visiting locations associated with alcohol consumption, such as bars, pubs, nightclubs, and liquor stores, on domestic violence at a neighborhood level [11].

The integration of geolocation data with environmental exposure datasets has emerged as a vital area in environmental health and epidemiology, particularly in the context of spatial energetics. The concept of spatial energetics highlights the revolutionary ability to examine dynamic, high-spatiotemporal resolution data on location and time-matched energetics through GPS, accelerometry, and GIS [12]. This approach has allowed researchers to explore how environmental characteristics, space, and time are linked to activity-related health behaviors, such as obesity and physical inactivity. Another article further investigated the spatial physical activity patterns among children in different socioeconomic neighborhoods, emphasizing the importance of safe streets and access to recreational facilities [13]. Moreover, GPS-based activity space measures of environmental exposure, such as walkability and greenness, were associated with increased physical activity [14]. Soon after the start of the COVID-19 pandemic, it became evident that geolocation data would play an increasing role in public health, statistics, and disease outbreaks. Companies such as Safegraph began partnering with academic institutions, and a large number of high-impact manuscripts started being published showing the contributions of geolocation data to public health [11]. These studies collectively underline the potential of geolocation tracking data in linking environmental exposures such as air pollutants, NDVI, and walkability to health outcomes. However, they also highlight the need for addressing challenges related to technical problems, statistical methodologies, participant privacy, and security.

Tracking participants is challenging in many contexts, including healthcare studies and large-scale surveys. One of the biggest challenges is ensuring the security and privacy of the data collected. Participants may be concerned about who can access their personal information and how it will be used. Another challenge is power constraints, especially for precise location monitoring, which requires GPS receivers. GPS receivers consume a relatively large amount of power; thus, over-using or keeping them on for long periods will significantly affect the battery life of smartphones [15]. This can be especially problematic when tracking participants in longitudinal health studies over extended periods. Previous apps, such as Beiwe [16], bear similarities to this contribution, but did not focus solely on collecting geolocation data or evaluated the quality of the data collected. Another similar app called CareConekta [17] focused on the feasibility of using an app to collect GPS location. This app only collected two GPS heartbeats per day and the location accuracy was approximately 1km; however, our app continuously collects precise GPS data while the user is moving. Other studies have attempted to evaluate the accuracy of geolocation data collected from smartphones but they did not account for energy consumption or private data handling [18,19]. Finally, software scalability is another major issue when tracking large numbers of participants. As the number of participants grows, managing and analyzing the data collected becomes increasingly difficult. Addressing these challenges requires combining technical and organizational solutions, such as improving data encryption and security measures, developing more efficient tracking algorithms, and implementing effective data management strategies that can handle large volumes of data.

We present a design for a mobile phone application that aims to utilize advances in modern smartphone software and hardware to create a reliable tracking solution for researchers interested in collecting geolocation data. Researchers can use the collected data to analyze the trajectories of patients and identify patterns that can help predict health outcomes. Our design considers the previously mentioned constraints and attempts to create a private and efficient way of collecting geolocation data for use in research. Our contributions are:Battery-efficient tracking logic that uses smartphone sensors to monitor physical activity and collect geolocation data.Scalable cloud architecture for sending and storing geolocation data while managing cost.Using Mobile Ads ID (MAID) to ensure data privacy, integrity, and usability in healthcare research.

## 2. Materials and Methods

This section outlines the development of an app designed to record users’ locations, focusing on both accuracy and energy efficiency. A novel algorithm is implemented to switch the GPS receiver’s power on and off based on the phone’s motion status, ensuring continuous tracking while conserving battery life. In addition, we describe a cloud-based database architecture to securely store large volumes of data and allow authorized entities to access it for analysis. Finally, we present a description of a built-in rewards system that utilizes a payment Application Programming Interface (API) and cryptocurrencies. Figure 1 shows a summary of the methods.

### 2.1. Energy-Efficient Geotracking Logic

The primary purpose of the app is to record the location of subjects. It is always essential that location tracking works when the user is moving. The collected trajectory could be used in location-based services such as matching the location data with Points of Interest (POIs) data to obtain a history of locations that the user visited. The history is used to research relationships between visiting specific locations and health outcomes; for example, visiting HIV clinics can help predict if the user has a sexually transmitted disease [20]. Gaps in the trajectory will lead to inaccurate measures of the visits; thus, the missing behavioral data points about users [21,22].

Another important requirement when it comes to collecting location records is accuracy. Modern smartphones contain GPS receivers that can detect location with 1–10 m accuracy via measuring the arrival time of signals from four or more satellites that orbit the Earth [23]. Latitude and longitude from GPS are often assisted with WIFI and cellular tower locations, known as Assisted GPS (AGPS). WIFI location is derived from nearby networks and can improve location accuracy (combined with GPS) indoors [24]. Cellular tower locations rely on triangulation between cell phone towers and have the lowest accuracy of all sensors [25]. Table 1 shows the accuracy of different sensors in smartphones. WIFI and cellular location can only narrow down location to 10 m. This needs to be more precise as POIs can be smaller and thus require more precision in the location data. However, GPS receivers use a large amount of energy, which is often a limitation for smartphones. In Table 1, low energy uses refers to sensors that have minimal impact on the device’s battery life, allowing it to last for several hours or even days of normal use without requiring a recharge. High energy use, on the other hand, refers to sensors that have a more substantial impact on the device’s battery life and might noticeably decrease the battery life of the device [26]. With high-performing CPUs and memories, energy-optimizing techniques are implemented in smartphones to balance the different needs of the user. GPS receivers consume significant power when they actively receive and process satellite signals. By turning the GPS receiver on and off based on usage, smartphones can optimize their battery usage and extend the time between charges. When an app or system service requires location information, the GPS receiver can be temporarily activated to provide the needed data and then deactivated once the location has been determined [27].

Given the energy constraints and the need to track users constantly, we use other phone sensors such as motion detection, accelerometer, gyroscope, and magnetometer to meet the requirements. These sensors use significantly less energy than GPS [28], and they can be used to implement a logic to switch the power of the GPS receiver based on the motion status of the phone. The Algorithm 1 shows how we use these sensors to optimize energy consumption while always tracking.
**Algorithm 1.** Enhanced Motion-Assisted GPS Location Tracking1. Initialize the tracking system2. Set *t*_0_ = current time3. While (tracking is not terminated):      3.1. Query *movement*_*status* from the Operating System      3.2. If *movement*_*status* = *STATIC*:           3.2.1. Wait for Δ*t* = 5 min           3.2.2. Deactivate GPS data requests           3.2.3. Monitor for changes in *movement*_*status*      3.3. Else if *movement*_*status* = *IN*_*MOTITION*:           3.3.1. Activate GPS data requests           3.3.2. Store GPS data in *GPS*_*location*_*data*           3.3.3. Monitor for changes in *movement*_*status*      3.4. Set *t*_0_ = current time

Our algorithm is based on the React Native Background Geolocation framework [29]. The framework supports distance-based filtering of location in iOS. Similarly, our algorithm uses distance-based filtering as a standard for both iOS and Android, to ensure similar data quality in both platforms. Motion sensors can detect different motion statuses such as stationary, walking, driving, etc. When motion changes from static to another state, we start the tracking with the highest possible accuracy (AGPS). We then filter out locations based on 5 m intervals. Motion detection is performed differently based on the operating system of the phone. The proposed algorithm is efficient because it does not use energy when the mobile device is static. Once the movement status changes, the app will start requesting data from the GPS receiver sensor.

For iPhones, motion detection is performed by the M-series chips (M7, M8, M9, and M10), also known as motion coprocessors. The motion coprocessor’s function always collects and processes sensor data, even when the iPhone sleeps. Motion coprocessors are used to keep track of the motion status and physical activity without engaging the main processor, thus, conserving energy by ensuring high-energy consumption hardware components (such as GPS receiver and processor) are only used when users move. The motion status of the phone is constantly being monitored using the CMMotionActivityManager API [30]. Once the motion status is changed from static to in-motion, the algorithm will start utilizing GPS and record location, and vice versa. After being static for a long time, iOS requires the phone to move outside a “stationary geofence” (approximately 200 m). iOS has more restrictions in the limitations section of the discussion. Android OS employs a similar technique for monitoring the motion status and physical activity using ActivityRecognitionClient API [31]. However, the hardware components vary between different manufacturers.

We also account for other edge cases that might halt the tracking due to OS restrictions. These cases include terminating the app or restarting the phone. iOS and Android allow apps to ask user permission to fetch location even when terminated. In iOS, this is achieved via BGAppRefreshTask [32], which periodically fetches data and performs small tasks in the background. In Android OS, the app operates by starting a foreground service [33]. This means the app can continue monitoring motion status and fetch geolocation at the same rate as before termination. It becomes immune to the OS, terminating it to free up resources. Moreover, apps can also start automatically in the background after the phone restarts without action from the user. iOS still requires the phone to move approximately 200 m after the restart to resume tracking.

### 2.2. Cloud Architecture

Geolocation data have high frequency and volume and are very sensitive as they can be used to infer personal addresses and workplaces. While data are collected via the app, a scalable cloud architecture is crucial to handle the data safely and securely. Our cloud architecture is built on Amazon Web Services (AWS), but it can be easily replicated using the alternative services on Google Cloud Platform (GCP), Microsoft Azure, or Cloudflare. Figure 2 illustrates the main components of our suggested cloud architecture of the app:

The first component of the cloud architecture is the Application Programming Interface (API). Our API (also referred to as the backend) receives geolocation data from the app and writes it into the database. Due to the large volume and frequency of geolocation data, geolocation data are sent in batches of 250 points. Each point contains latitude, longitude, altitude, and accuracy. If the batch is successfully recorded into the database, the API will send a success message to the app (also referred to as frontend). The API will send an error message to the app if an error happens. The data that failed to be written into the database will be saved in the app locally in an SQLite database. The app will retry to send the data records at a later time to ensure no records of the geolocation data are lost. Possible reasons for the failure of sending data could be loss of internet connection (from the phone side) or high load on the API server or database.

We use Domain Name Service (DNS) and Hypertext Transfer Protocol Secure (HTTPS) protocols which play a crucial role in protecting the privacy of the transmitted data over the internet. DNS-over-HTTPS (DoH) adds an extra layer of encryption to DNS queries, preventing eavesdropping and tampering [34]. HTTPS, on the other hand, secures data transmission between the user’s browser and the website server by encrypting the data with Secure Sockets Layer (SSL) certificates [35]. By combining the secure features of both protocols, users can maintain their privacy and reduce the risk of cyberattacks or unauthorized access to their sensitive information.

Furthermore, our implementation of load-balancing and autoscaling groups greatly enhances the scalability of our API, effectively addressing potential challenges posed by a large user base. When the server experiences an increase in load due to a high number of users, the effective use of autoscaling groups can dynamically adjust server capacity to accommodate this demand. The load balancing, on the other hand, distributes incoming traffic evenly across multiple servers, preventing any single server from becoming overwhelmed. This ensures optimal performance and fault tolerance if a server fails or experiences issues. Together, load-balancing and autoscaling groups enable the system to handle increased traffic seamlessly and optimize resource utilization. This translates to cost efficiency, as we can balance server capacity and operational costs. Instead of consistently running many servers, which can be expensive, we can dynamically allocate resources as needed, ensuring that we only pay for the capacity we require.

We use AWS Dynamo DB, a non-SQL scalable serverless database. Using a non-SQL database provides flexibility for storing different data formats. This is helpful because geolocation data records might be formatted differently across different platforms (iOS/Android). Moreover, using a serverless database reduces the cost of storing data on the cloud compared to traditional SQL databases. Traditional databases consume fixed computational resources allocated to the server that executes queries. On the other hand, serverless databases provide a cost-effective and efficient solution for managing high-volume requests. Unlike traditional database solutions, DynamoDB charges only for the actual number of reads and writes, reducing costs during periods of low query activity. This pay-per-use model ensures that we only pay for the resources we consume. During periods of high request volume, DynamoDB is designed to automatically scale its capacity to meet the increased demand, alleviate pressure on the database and maintain consistent performance. In addition, DynamoDB has scalable storage space. The storage cost is based on consumption; thus, the cost is only based on the volume of data stored in the database.

The third component of the cloud architecture is automated remote notification jobs. The purpose of this component is to send notifications to remind and encourage users to take the necessary actions needed to ensure data collection for the study. Notifications are typically sent based on a predetermined schedule, which could be daily, weekly, or at other specific intervals, depending on the requirements of the study. Algorithm 2 illustrates the logic used in automated notifications.
**Algorithm 2.** Automated Remote Notification for User Engagement1. Query user IDs and the last notification time from the database and store it in user_information_list2. For Each User *u in* user_information_list:      2.1. Query number of events that belong to *u* and store it in *events*      2.2. For event *e* in *events*           2.2.1. If *e.event_type* = *PERMISSIONS_UPDATE*                2.2.1.1. If *e.values*
**=**
*true*                     2.2.1.1.1. Remove user *u* from notification table                2.2.1.2. If *e.values* = *false*                     2.2.1.2.1. Set *t*_0_ = current time                     2.2.1.2.2. Set *t*_1_ = last notification time for the user                     2.2.1.2.3. If Δ*t* ≥ user’s notification interval:                          2.2.1.2.3.1. Set *t* = *u.token*                          2.2.1.2.3.2. Send notification using Firebase                          2.2.1.2.3.3. Update the database with the new notification information (i.e., date and time sent)

To automate the execution of the algorithm and send remote notifications, we use Apache Airflow, an open-source platform for orchestrating complex workflows. Airflow enables us to schedule, monitor, and manage the remote notification algorithm more efficiently and reliably.

Lastly, the automated jobs for rewards aim to send incentives to users’ accounts upon completing their participation in the studies. The incentives are in the form of cryptocurrencies or payments made through payment APIs in exchange for the data contributed by users. Rewards are also automated using Apache Airflow. Algorithm 3 illustrates the logic used to compensate users.
**Algorithm 3.** Automated Rewards for user compensation1. Query user IDs and balance from the database and store it in user_information_list2. For Each User *u in* user_information_list:      2.1. Query number of events that belong to *u* and store it in *events*      2.2. For event *e* in *events*           2.2.1. If *e.event_type* = *START_TRACKING*                2.2.1.1. Set *t*_0_ = current time                2.2.1.2. Set *t*_1_ = *e.timestamp*                2.2.1.3. Set *t*_2_ = number of days required by the study                2.2.1.4. If *t*_0_ − *t*_1_ ≥ *t*_2_                     2.2.1.4.1. Obtain the user-preferred payment method                     2.2.1.4.2. Update user balance                     2.2.1.4.1. If user preferred payment method is CYPTO                          2.2.1.4.1 Execute smart contract                     2.2.1.4.2. Else                          2.2.1.4.1. Call PAYMENT_API

### 2.3. Data Quality and Privacy

Given that the app’s primary purpose is collecting geolocation data for research, it is crucial to design a data schema that ensures ease of access, scalability, and privacy protection. An effective data schema indexes geolocation data using latitude and longitude, a method widely used in Geographic Information Systems (GISs) and recommended by data management practices for geolocation data [36]. This approach facilitates efficient retrieval and analysis, seamlessly integrating the collected data with other geospatial datasets or tools [37].

The schema should separate personally identifiable information (PII) from geolocation data to preserve user privacy. This practice is in line with the privacy by design principle outlined in the General Data Protection Regulation (GDPR), and it is achieved using Mobile Advertiser IDs (MAIDs), an anonymized identifier unique to each device, linking the PII and geolocation data [38]. Through this design, researchers can access the geolocation data without directly exposing any sensitive information, thereby ensuring user privacy is maintained following established privacy standards [39]. Table 2 shows the proposed database schema.

The design follows the star schema, widely used due to its simplicity, consistency, efficiency in handling large datasets, and ease of integration with other data sources. The central table in the schema is the User Metadata. Each user/device has one row in that table which stores the Mobile Advertising ID (MAID). MAID is used as a foreign key in most of the other tables to ensure that the data are easily accessible to researchers. The Geolocation Records table contains all the collected location data and can be indexed using the coordinates to increase the efficiency of spatial queries [40]. The Events table includes status updates from the app that are used internally to keep track of the study progress and the status of the permissions provided to the app. The Rewards and Notifications tables keep track of the data needed to run the automated jobs to compensate users and send remote notifications in case any actions are needed. Finally, the Studies table includes metadata that are presented on the User Interface (UI).

Using the MAID as the anonymized identifier for geolocation data collection in an app offers several advantages. The mobile operating system generates MAIDs and they are unique to each device, ensuring a consistent and distinct identifier for each user. This uniqueness is essential for accurately linking geolocation data with user-specific information while maintaining anonymity. The app prompts users to share their MAID according to the latest privacy updates on iOS and Android using a framework designed for this use case [41].

### 2.4. User Engagement and Usability

To ensure that the app collects the data required for the studies, it is crucial to have a friendly user experience [42]. A friendly user experience encompasses several key elements. Firstly, it is important to ensure that the app requests the necessary permissions to collect location data [43]. This involves making it easy for users to grant these permissions with a transparent explanation of why the app needs them. By being transparent and providing a rationale, users may be more likely to feel comfortable sharing their location data. Secondly, the app should keep users informed about the status of the studies [44]. This can be achieved by providing real-time updates on the progress of the studies, including the number of days left for data collection and any milestones reached. This information is displayed to ensure that users are always aware of how their data contributes to the research. Lastly, to preserve user privacy and comfort, the app must offer an option for users to disable location tracking as needed. This feature should be easily accessible and visible within the app, allowing users to maintain control over their data and build trust in the app’s commitment to respecting their privacy. Figure 3 shows the features that were taken into consideration when designing the UI.

Figure 4 shows the UI design of the app. More details are shown in Appendix A.

After signing into the app, the user is presented with four pages. First, the Dashboard shows the progress of the studies in which the user is enrolled. Second, the Data page is where the user can view metrics on the geodata collected; for example, they can visualize the data on a map. Third, the Rewards page lets the user view previous or pending compensations. Fourth, the Settings page is where the user can view all given or pending permissions required by the app to collect geolocation data and exit the study.

### 2.5. Rewards System

#### 2.5.1. Blockchain

The first method of rewarding participants for sharing their geolocation data is blockchain. Using smart contracts on a decentralized platform, users can receive incentives from Non-Fungible Tokens (NFTs) or cryptocurrencies. When users share their geolocation data with a research app, this system triggers a smart contract, automatically verifying the information. Previous studies have highlighted that blockchain can provide a secure environment for sensor data exchange, particularly in Internet of Things (IoT) networks, due to its cryptographic validation processes [45]. Moreover, another study has shown how blockchain can incentivize data sharing from vehicle sensors in a transparent and tamper-proof manner [46].

Using blockchain-based smart contracts for rewarding geolocation data sharing comes with several advantages. Firstly, it has faster processing times, especially for compensating international participants who might not be able to receive bank transfers. International bank transfers are also subject to transfer rates that might change during the study period and lead to an increase in the study budget or a decrease in the compensation received by the user [47]. Secondly, it creates secure and anonymous transactions, giving users confidence in the system’s integrity and reducing the PII provided to the app. Furthermore, cryptocurrencies can be easily exchanged for other digital assets or physical currencies, increasing the utility of the incentives [48].

However, there are also potential drawbacks to consider. One concern is the variable fees associated with sending payments using the blockchain. Fees might vary depending on the cryptocurrency’s price and time of the day. Additionally, the fluctuating value of cryptocurrencies may deter some users from participating due to concerns about the stability of their rewards [49]. Moreover, users might not be familiar with blockchain or might not trust in their value. The participants need to own a cryptocurrency wallet to receive compensation in the form of NFTs or cryptocurrencies. Additionally, participants need a stable internet connection and knowledge of how to accept and spend their digital assets. Lastly, the deanonymization of public blockchain transactions could lead to others knowing the identity of the study participants [50].

#### 2.5.2. Payment APIs

Another alternative is integrating payment APIs into the app, which can provide a seamless and efficient way to reward users for sharing their geolocation data. Payment APIs facilitate secure and instant transactions between the app and its users. The implementation of payment APIs in research apps streamlines the rewarding process. Once users share their geolocation data, the backend system verifies the information and triggers a payment transaction through the chosen payment API. The API then communicates with the payment processor to complete the transaction, transferring the rewards directly to the user’s bank account.

Payment APIs automatically distribute rewards quickly and efficiently. Users can select their preferred payment methods, which enhances convenience and encourages participation. It also reduces researchers’ time and effort to hand out the rewards.

However, there are potential disadvantages to consider. Transaction fees may be incurred when using payment APIs, which could impact the overall budget for user rewards. Moreover, the app’s dependency on third-party payment processors introduces potential risks, such as service disruptions or changes in fee structures, which can impact the stability of the reward system.

## 3. Results

The app was installed on an iPhone 14 Pro Max and tested for 20 consecutive days. Next, the collected data were retrieved from the database by querying all records with matching email and MAID. Figure 5 shows the total distance traveled by the phone owner. Moreover, we have used the activity recognition field which describes the state of the phone holder (still, walking, in vehicle) to calculate the valid tracking minutes per day shown in Figure 5. Valid tracking minutes per day are defined as the sum of the time difference between consecutive timestamps when the phone’s motion status is in motion.

### 3.1. Accuracy of Geolocation Data

The user’s location data are overlayed on the OpenStreetMap (OSM) data, a community-driven mapping platform known for its detailed, up-to-date spatial datasets. Each geolocation point is assigned an ‘indoors’ or ‘outdoors’ classification depending on its position relative to the footprint of buildings mapped on OSM. This classification is carried out through a computational geometry method known as point-in-polygon, where each point is checked to see if it lies within the polygon that represents a building. A total of 6% of the data received from the database was indoor data. We then employ the location accuracy metadata provided by the iPhone’s built-in GPS and location services to assess the precision of indoor vs. outdoor points. Table 3 compares the accuracy of collected geolocation data in indoor versus outdoor settings. As seen in Table 1, outdoor accuracy was on average 8.29 m, which is slightly higher than the maximum GPS accuracy (5 m). The difference can be due to signal blockage by buildings or trees and atmospheric conditions. Indoor accuracy had an average of 30.6 m, with the 25th percentile at 3.1 m, the 50th percentile at 14.2 m, and the 75th percentile at 35 m. The difference in accuracy can be largely attributed to the variability in WIFI and cellular service strength, impacting the precision of the location data.

### 3.2. Activity Recognition

Next, we evaluated the proficiency of the activity recognition algorithm in detecting and tracking user movement. Considering that the tracking algorithm automatically ceases to record data after five minutes of inactivity, our team focused on identifying gaps in the data—instances where timestamps were more than five minutes apart. These gaps imply periods of presumed inactivity, followed by the recommencement of movement. To investigate the algorithm’s responsiveness further, we measured the geographic distance between the last recorded location before the tracking was paused and the first location registered when it resumed. This allowed us to assess how promptly the algorithm initiates tracking once the user begins moving again. The distances the user moves before the algorithm resumes tracking are plotted in Figure 6. As shown in Figure 6, when our tracking stops and restarts again, the user has moved less than 200 m. That confirms that the tracking restarts within the stationary geofence of iOS as described in the methods section.

### 3.3. Battery Consumption

Moreover, we recorded the daily battery usage (in percentage) and background activity (in minutes) from the iPhone Settings app. Figure 7 shows the recorded battery usage values. As shown in Figure 7, the number of hours reported by the iPhone Settings App was consistent with the number of valid tracking minutes reported in Figure 5.

## 4. Discussion

### 4.1. App Novelty in the mHealth Field

Our app introduces an energy-efficient tracking feature, allowing for continuous and seamless monitoring of users’ geolocation data without draining the battery life of their devices. By employing intelligent algorithms and state-of-the-art energy management techniques, this innovative approach ensures that researchers can gather accurate and extensive data while minimizing the impact on users’ daily activities.

In a novel application of MAID, our app harnesses the power of big data and advanced machine learning techniques to analyze geolocation data and identify patterns of human interactions in various settings. This innovative approach enables researchers to better understand disease transmission dynamics, social networks, and human behavior in relation to health outcomes.

In addition, our app integrates blockchain technology to offer a streamlined rewards system for participants and incentivize their engagement in research studies. By leveraging the tamper-proof nature of blockchain, our app ensures that users are fairly compensated for their contributions while maintaining the integrity of research data. This reward mechanism encourages participation and could be a new way to crowdsource a valuable data source, ultimately enhancing the quality and impact of mHealth research.

Moreover, the app features a robust and user-friendly software tool to facilitate geolocation data collection, analysis, and visualization for research purposes. This comprehensive platform allows researchers to easily manage large datasets, generate insightful reports, and extract meaningful information to inform public health decision making.

### 4.2. Use Cases of Geolocation Data in Health

Geolocation data could be a rich source of insights for researchers in healthcare, public health, and other fields. When combined with other data sources, geolocation provides information about behavioral activities, socio-economic status, and exposure to certain activities. For example, POIs data (location name, NAICS code, polygon shape, and working hours) infer the activities performed in certain locations. Using geocoding techniques, POIs and geolocation data can be combined to obtain a history of the locations visited during the day, also known as the trajectory. Figure 8 shows an application of geolocation data in analyzing the relationship between trajectories and health outcomes using a pipeline for merging POIs and geolocation data to generate the trajectories.

In the depicted process, the initial step involves integrating geolocation information with POIs using geofencing or point-in-polygon overlay techniques [51]. Subsequently, the locations visited are merged with medical health records or health outcomes documented during a study. This approach can be further developed to identify high-risk individuals requiring medical interventions. Previous studies have shown promising results using similar methods to study human behavior and health outcomes using geolocation data [7,52,53].

The utilization of geospatial methods in behavioral research, particularly in the context of public health, could be extended beyond simple overlaying of geolocations with POIs using the point-to-polygon method. The scikit-mobility library provides a comprehensive framework for analyzing and simulating human mobility, encompassing collective and individual human mobility models such as gravity and radiation models, and the spatiotemporal clustering algorithms which help uncover visit durations, frequency and accuracy [54]. Furthermore, integrating socioeconomic datasets, such as Census, allows researchers to explore environmental features and contribute to a more sophisticated understanding of geolocation data in public health research [55,56].

Another potential application of the data gathered by our software involves utilizing MAIDs to combine geolocation data with other data sources while ensuring that no personally identifiable information is disclosed. For instance, MAIDs can facilitate the collection of details about online behavior, including purchases, browsing history, social media usage, and other interests. Additionally, MAIDs can send surveys or advertisements to study participants via mobile devices. This allows a streamlined method of gathering information from individuals located remotely and providing health interventions using advertisements.

### 4.3. Ethical Considerations for Using Geolocation Data in Public Health Research

Even with data privacy measures and encryption technologies in place, the privacy concerns of using geolocation data in research are still an active research topic. Several methods have been used to protect the privacy of data and address the related ethical concerns. For example, deidentified geolocation data have been widely adopted to ensure that individuals’ personal information is protected [57,58,59,60,61]. Moreover, representative datasets from diverse groups and geographical locations are crucial in using mobility data for healthcare research [62,63,64,65]. Collecting larger datasets using this app will allow for more studies to utilize geolocation data to investigate the effects of mobility on health outcomes and reduce the barriers, effort, and time needed by different groups to participate in research. In the proposed application, we ensure that users know the purpose and methods used in the studies by showing the users the student consent app before collecting geolocation data. Moreover, the app provides data visualization features so the users can keep track of what data have been collected about them.

In the context of research, public health researchers, who utilize this app to gather research data, are responsible to inform subjects of potential effects and ethical considerations, especially for individual users. This app, like the ones used for contact tracing during the COVID 19 pandemic or personalized exposure assessment, provides opportunities to collect real time data on movement, health status, and environmental exposures [66,67]. Certain features have been implemented in the app to ensure that the users are informed about potential risks of using the app, such as the study’s consent form and ability to stop tracking temporarily. However, the decision to implement apps for data collection needs consideration of the potential risks and benefits for individual users [68]. The impact on individuals may involve increased surveillance and control measures, potential privacy risks, and the chance of misinterpreting data which could lead to unintended harms to data owners [69]. Ethical considerations should also be a priority in terms of transparency in app usage plans, decision-making processes regarding implementation, informing individuals about risks or benefits they may face, as well as measures to ensure confidentiality and consent when handling collected data [70]. Given the nature of data collection through apps, it is crucial to establish a strong ethical framework that respects individual autonomy while maximizing technology’s potential benefits [71].

### 4.4. Limitations and Future Work

There are limitations with this paper that warrant discussion. The most prominent of these limitations is connected to the architecture of the iOS operating system itself. For privacy and battery life considerations, iOS is designed to control background processes strictly. Consequently, our application cannot run background jobs when the user has terminated it, making it impossible to verify if tracking services are active without user intervention consistently. Another limitation is the accuracy of geolocation data indoors, which can lead to misattributed POI visits. Error in the location coordinates must be considered in the analysis phase using lower geocoding constraints or probabilistic visits attribution. Future applications of our methodology must include a detailed analysis of data quality, specific to the group’s characteristics and the cohort in question. For instance, if applied to a cohort with depression and limited mobility, the analysis may focus on indoor geolocation tracking accuracy, consider variations in smartphone technology, and tailor additional analyses to factors such as different socioeconomic statuses or physical activity levels. In addition, the data presented in the Section 3 were collected from an iOS device, which might not be representative of the diverse range of operating systems and hardware configurations. Future research will be required to explore the discrepancies and effects across various operating systems and devices, as the current test may have neglected differences that could influence the functionality and results. Comparing the data collected by the app to the subjects’ self-reports would also strengthen this work.

In light of the limitations outlined above, there are several potential directions for future research and development to explore. A key aspect of our future work will be the focus on enhancing the privacy features of our application. One strategy we are considering is the implementation of Advanced Encryption Standard (AES) for location data. AES is a symmetric encryption algorithm that has become the industry standard for data security. Applying this encryption to location data would provide robust protection against unauthorized access and significantly enhance the system’s privacy.

## 5. Conclusions

Passive data collection of geolocation data using mobile devices has the potential to revolutionize healthcare. It provides valuable insights into patient health and behavior, allowing for an in-depth understanding of public health epidemics, including substance abuse and sexually transmitted diseases. Using personal devices as a source of new datasets, healthcare researchers could unlock opportunities to study their subjects’ daily activities without intervening or reducing study budgets. Geolocation data could allow the study of socioeconomic and behavioral factors on health outcomes. Integrating health records with geolocation data allows researchers to examine the relationships between the type and frequency of locations visited and their effect on the patient’s health condition. This method allows researchers to tap into the geographical contexts that may interfere with or facilitate receiving healthcare services. Moreover, geolocation data could also facilitate the discovery of patterns linking similar health outcomes in patients who frequent similar locations, thus enabling a closer identification and understanding of their specific needs. Although still in its infancy, research using geolocation data holds promise in being able to improve the surveillance and delivery of public health and medicine rapidly and profoundly.

## Figures and Tables

**Figure 1 sensors-23-07917-f001:**
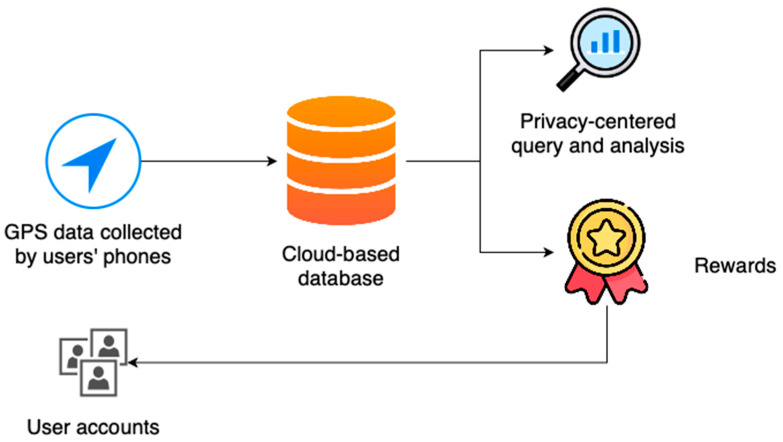
Summary of the methods.

**Figure 2 sensors-23-07917-f002:**
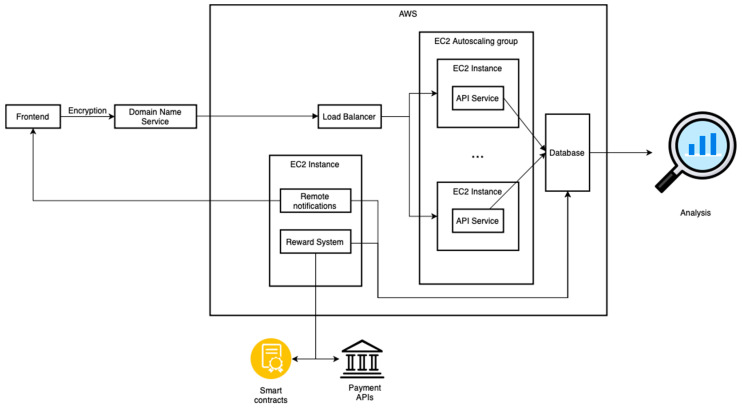
Cloud architecture of the app.

**Figure 3 sensors-23-07917-f003:**
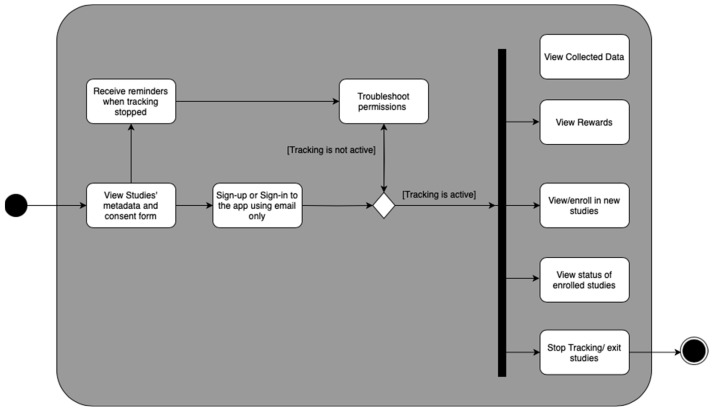
Activity diagram of the required features for the app.

**Figure 4 sensors-23-07917-f004:**
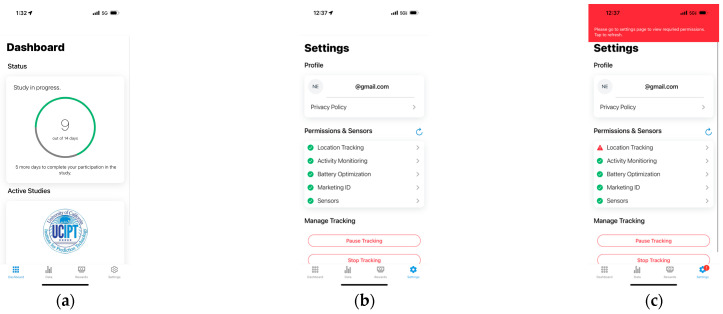
Screenshots from the app UI: (**a**) Dashboard showing the status of the study and active studies that the user is enrolled in; (**b**) Settings page that provides feedback on the needed permissions, privacy policy, and options to pause tracking or exit the study; (**c**) example of error messages when required permissions are not granted along with action items for the user to complete to ensure that their data are being collected.

**Figure 5 sensors-23-07917-f005:**
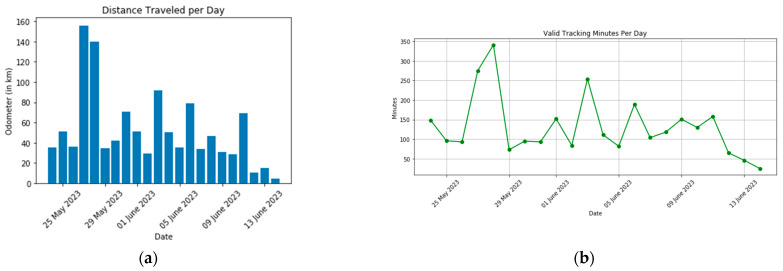
(**a**) Distance traveled each day as captured by the app; (**b**) valid tracking minutes per day.

**Figure 6 sensors-23-07917-f006:**
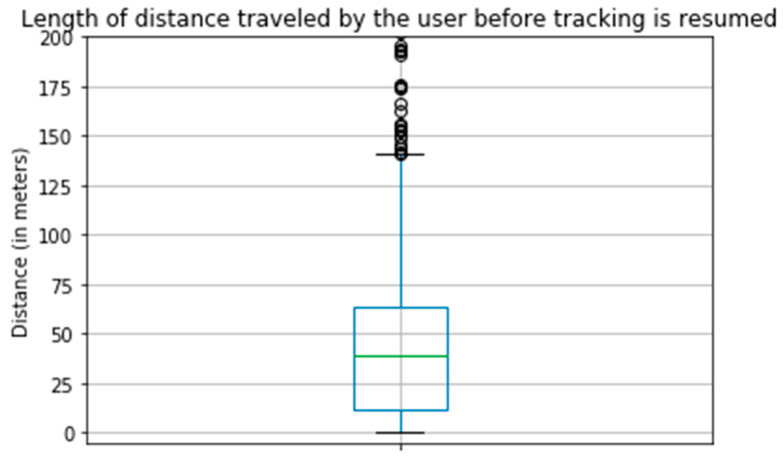
Length of distance traveled by the user before tracking resumed.

**Figure 7 sensors-23-07917-f007:**
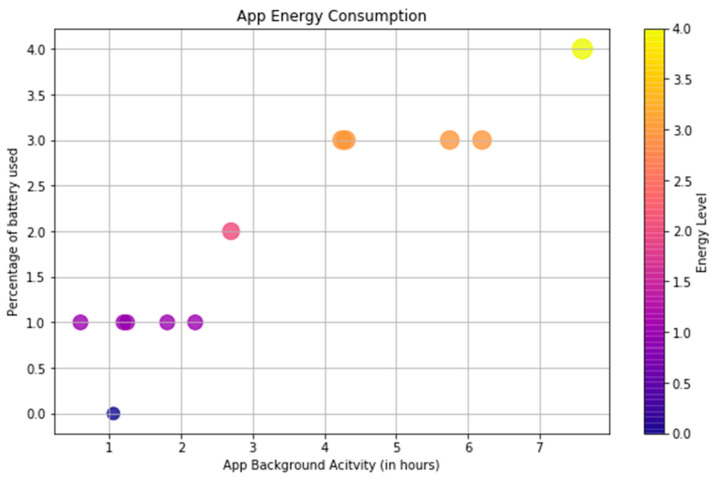
Energy consumed by the app as reported in the iPhone Settings app.

**Figure 8 sensors-23-07917-f008:**
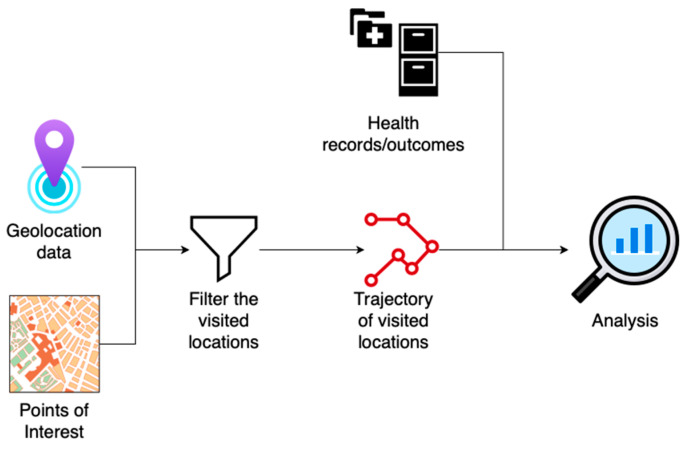
Geolocation data processing and analysis pipeline: a use case of geolocation data in public health.

**Table 1 sensors-23-07917-t001:** Comparison between accuracy and energy consumption of different location sensors in smartphones.

Sensor	Accuracy	Energy
Cellular	100 m	Low
WIFI	10–50 m	Low
GPS	5–10 m (outdoors) 10–20 m (indoors)	High

**Table 2 sensors-23-07917-t002:** Database schema.

Table	Column	Description	Example
User Metadata	MAID	Mobile Advertising ID, unique identifier for each phone (primary key)	C3F0267B-CA56-4541-929D- E2BAC4979AC2
	Email	Email address of the user	
	Phone Metadata	Operating system type and version number	
	IP Address	Internet Protocol Address	
	Firebase Token	Unique token for each device	
Geolocation Rewards	UUID	Universally Unique ID for each location (primary key)	964D5405-BD8D-4E9C-A221-442C5E92ED81
	MAID	Unique identifier for each phone (foreign key)	
	Study ID	Study Code (foreign key)	
	Timestamp	Time, data, and the time zone when location was recorded	
	Activity	The type of motion detected when location was recorded	still, on_foot, walking, running, in_vehicle, on_bicycle, unknown
	Battery	Percent of battery charge when the location was recorded	
	Coordinates	Latitude, longitude, altitude, and accuracy of the location	
	Odometer	Distance moved	
Events	UUID	Universally Unique ID for each event (primary key)	
	MAID	Unique identifier for each phone (foreign key)	
	Event Type	Code that corresponds to internal use	PERMISSIONS_UPDATE, TRACKING_STARTED
	Values	Metadata about the event, such as timestamp or Boolean values	
Studies	Study ID	Study Code (primary key)	
	Metadata	Description of the study, start/end dates, and other requirements	
	MAID	Unique identifier for each phone (foreign key)	
Notifications	Timestamp	Time when row was last updated	
	Notification Type	Code that corresponds to internal use	
	MAID	Unique identifier for each phone (foreign key)	
Rewards	Balance	Value of rewards sent to the user	

**Table 3 sensors-23-07917-t003:** Distribution of the geolocation data accuracy reported in iPhone location metadata.

	Indoor	Outdoor
Number of data points	1415	24,191
Mean	30.6 m	8.29 m
Standard Deviation	33.58	109.44
25th percentile	3.1 m	4.7 m
50th percentile	14.2 m	4.7 m
75th percentile	35 m	4.7 m

## Data Availability

Data and code are available on request due to restrictions.

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
