# Peer review of "A Mobile Health Application Using Geolocation for Behavioral Activity Tracking"

_sensors, 2023, doi:10.3390/s23187917_

Round 1

Reviewer 1 Report (New Reviewer)

This manuscript is a valuable addition to the existing literature on geolocation tracking for health behavior research. Despite its merits, I have three major concerns:

Major comment #1

Lit review: the authors have overlooked a large body of environmental health and epidemiology literature that links geolocation to environmental exposure datasets (air pollutants, NDVI, and walkability) in space and time. The authors seem to imply this in the Introduction and Conclusions sections, but this should be discussed more explicitly as this is one of the most common applications of geolocation tracking data on health, an area referred to as Spatial Energetics by James et al:

James, P., Jankowska, M., Marx, C., Hart, J.E., Berrigan, D., Kerr, J., Hurvitz, P.M., Hipp, J.A. and Laden, F., 2016. “Spatial energetics”: integrating data from GPS, accelerometry, and GIS to address obesity and inactivity. American journal of preventive medicine, 51(5), pp.792-800.

Similarly, the authors’ understanding of geospatial methods in using geolocation data for behavioral research also seems somewhat superficial. Yes, you can overlay geolocations with POIs using the point-to-polygon method to infer the context of each point. However, this is just a very simple way to do such public health research. Have we accounted for time spent at each location? Can spatial statistics be used to identify clusters of points to derive frequently visited locations? Geographically, buffers can also be created around the trajectories of geo-locations to account for the extent of environmental features such as noise and air pollutants. The literature review by Yi et al. contains excellent visual representations (Fig. 3) of how various spatial methods can be used to link geolocation to health behaviors and answer various questions about health outcomes, which maybe of help for the authors.

Yi, L., Wilson, J.P., Mason, T.B., Habre, R., Wang, S. and Dunton, G.F., 2019. Methodologies for assessing contextual exposure to the built environment in physical activity studies: A systematic review. Health & place, 60, p.102226.

Overall, the potential applications of the geolocation and behavioral data that the authors can collect with the app may have much broader potential than what they currently discuss in the manuscript, and this should be expanded as they seem to be leaving out the entire body of research.

Major comment #2

Why are you only using the iPhone to test your application? Since the application was developed for both iOS and Android systems, you acknowledged that there could be differences in how geo-tracking works in the two systems OS. Also, you seem to be using the latest version of the iPhone to test the app. Would not that be a bit biased considering that you will be installing the app in the future on study participants who may not have the latest smartphone. In my experience, some participants might even have a phone that is more than five years old. Also, who is testing the app? The first author? Would this person have a typical daily routine similar to the average person? What if we chose a person who is very active and constantly on the go, while future users are more sedentary and stay in their homes? Overall, the testing procedure seemed arbitrary and did not take into account many other factors.

Major comment #3

I am afraid that comparing indoor and outdoor accuracy does not mean much to me. What are you trying to say? I am not sure. To me, a performance result would at least mean that you are comparing some statistics (e.g. valid minutes per day, valid days per person, non-collection rates, etc.) I did not see anything here.

Also, in the activity detection algorithm, did the authors think to compare turning geolocation tracking on and off to the data with subject self-reports (i.e., the gold standard) to assess how well it performs? In my opinion, this would be a more important analysis to find out how well the activity recognition works, rather than simply creating a bar chart with the distributions of the length of distances.

Overall, I do not see any comparison between the performance of these apps and other apps developed for these purposes. For example, Onnela et al. at Harvard developed an app called Beiwe that is very similar:

https://www.hsph.harvard.edu/onnela-lab/beiwe-research-platform/

I would recommend that the authors do more thorough research on what is already out there to discuss the pros and cons of this app in more detail.

This is an overall well-written article. 

Author Response

Kindly view the attached PDF to view the response in the intended format.

Response to Reviewer 1 Comments

Point 1: Lit review: the authors have overlooked a large body of environmental health and epidemiology literature that links geolocation to environmental exposure datasets (air pollutants, NDVI, and walkability) in space and time. The authors seem to imply this in the Introduction and Conclusions sections, but this should be discussed more explicitly as this is one of the most common applications of geolocation tracking data on health, an area referred to as Spatial Energetics by James et al:

James, P., Jankowska, M., Marx, C., Hart, J.E., Berrigan, D., Kerr, J., Hurvitz, P.M., Hipp, J.A. and Laden, F., 2016. “Spatial energetics”: integrating data from GPS, accelerometry, and GIS to address obesity and inactivity. American journal of preventive medicine, 51(5), pp.792-800.

Similarly, the authors’ understanding of geospatial methods in using geolocation data for behavioral research also seems somewhat superficial. Yes, you can overlay geolocations with POIs using the point-to-polygon method to infer the context of each point. However, this is just a very simple way to do such public health research. Have we accounted for time spent at each location? Can spatial statistics be used to identify clusters of points to derive frequently visited locations? Geographically, buffers can also be created around the trajectories of geo-locations to account for the extent of environmental features such as noise and air pollutants. The literature review by Yi et al. contains excellent visual representations (Fig. 3) of how various spatial methods can be used to link geolocation to health behaviors and answer various questions about health outcomes, which maybe of help for the authors.

Yi, L., Wilson, J.P., Mason, T.B., Habre, R., Wang, S. and Dunton, G.F., 2019. Methodologies for assessing contextual exposure to the built environment in physical activity studies: A systematic review. Health & place, 60, p.102226.

Overall, the potential applications of the geolocation and behavioral data that the authors can collect with the app may have much broader potential than what they currently discuss in the manuscript, and this should be expanded as they seem to be leaving out the entire body of research.

Response 1: We thank you for your feedback and we have citied the following works in the introduction (Page 2, Paragraph 2) to discuss literature review on Spatial Energetics:

James P, Jankowska M, Marx C, et al. “Spatial Energetics”: Integrating Data From GPS, Accelerometry, and GIS to Address Obesity and Inactivity. Am J Prev Med. 2016 November;515:792–800. PMID: 27528538. https://doi.org/10.1016/j.amepre.2016.06.006.

Bürgi R, Tomatis L, Murer K, de Bruin ED. Spatial physical activity patterns among primary school children living in neighbourhoods of varying socioeconomic status: a cross-sectional study using accelerometry and Global Positioning System. BMC Public Health. 2016 March 22;16:282. PMID: 27000056. https://doi.org/10.1186/s12889-016-2954-8.

Marquet O, Hirsch JA, Kerr J, et al. GPS-based activity space exposure to greenness and walkability is associated with increased accelerometer-based physical activity. Environ Int. 2022 July;165:107317. PMID: 35660954. https://doi.org/10.1016/j.envint.2022.107317.

To address the point brought up by the reviewer about geospatial analysis methods, we added a paragraph to elaborate on some of the methods we use to analyze geolocation data (Page 14, Paragraph 3):

“The utilization of geospatial methods in behavioral research, particularly in the context of public health, could be extended beyond simple overlaying of geolocations with POIs using the point-to-polygon method. The scikit-mobility library provides a comprehensive framework for analyzing and simulating human mobility, encompassing collective and individual human mobility models such as gravity and radiation models, and the spatiotemporal clustering algorithms which help uncover visit durations, frequency and accuracy [51]. Furthermore, integrating socioeconomic datasets, such as Census, allows researchers to explore environmental features and contribute to a more sophisticated understanding of geolocation data in public health research [52,53].“

[51] Pappalardo L, Simini F, Barlacchi G, Pellungrini R. scikit-mobility: an open-source Python library for human mobility analysis and simulation. 2019 July 9.

[52] Cornacchia G, Pappalardo L. A Mechanistic Data-Driven Approach to Synthesize Human Mobility Considering the Spatial, Temporal, and Social Dimensions Together. ISPRS Int J Geo-Inf. 2021 September;109:599. https://doi.org/10.3390/ijgi10090599.

[53] Yi L, Wilson JP, Mason TB, Habre R, Wang S, Dunton GF. Methodologies for assessing contextual exposure to the built environment in physical activity studies: A systematic review. Health Place. 2019 November;60:102226. PMID: 31797771. https://doi.org/10.1016/j.healthplace.2019.102226.

Point 2: Why are you only using the iPhone to test your application? Since the application was developed for both iOS and Android systems, you acknowledged that there could be differences in how geo-tracking works in the two systems OS. Also, you seem to be using the latest version of the iPhone to test the app. Would not that be a bit biased considering that you will be installing the app in the future on study participants who may not have the latest smartphone. In my experience, some participants might even have a phone that is more than five years old. Also, who is testing the app? The first author? Would this person have a typical daily routine similar to the average person? What if we chose a person who is very active and constantly on the go, while future users are more sedentary and stay in their homes? Overall, the testing procedure seemed arbitrary and did not take into account many other factors.

Response 2: Although we initially submitted this manuscript as a theoretical framework/methods paper, during the last review it was requested that we include data. As it takes time to get approval and downloads through the Google and iOS stores, we responded to this request by providing data from iOS based on convenience. We agree with the reviewer that there are many potential differences and biases that may results from different operating systems.

We also agree with the reviewer that any subsequent paper that applies this methodology in a practical scenario should indeed include an analysis of data quality. The analysis should be specific to the characteristics of the group being studied and the actual cohort. For example, in a future study applying our methodology to a cohort of individuals suffering from depression with limited mobility, the analysis might include a careful examination of geolocation tracking accuracy in indoor settings, taking into consideration the common locations and movement patterns of this specific population. The analysis could also account for variations in smartphone models and operating systems, reflecting the real-world diversity of technology used by the participants. If the study were to include a broader group, different socioeconomic statuses, or varying levels of physical activity, additional analyses would be tailored to address those specific factors.

To address these comments and improve the science within the manuscript, we have added this issue in the limitations section as a limitation and need for future research focused more on data and results to address this issue.

“Future applications of our methodology must include a detailed analysis of data quality, specific to the group's characteristics and the cohort in question. For instance, if applied to a cohort with depression and limited mobility, the analysis may focus on indoor geolocation tracking accuracy, consider variations in smartphone technology, and tailor additional analyses to factors such as different socioeconomic statuses or physical activity levels. In addition, the data presented in the results section was collected from a single IOS device, which might not be representative of the diverse range of operating systems and hardware configurations. Future research will be required to explore the discrepancies and effects across various operating systems and devices, as the current test may have neglected differences that could influence the functionality and results. Comparing the data collected by the app to the subjects’ self-reports would also strengthen this work.” (Page 15, Paragraph 3)

Point 3: I am afraid that comparing indoor and outdoor accuracy does not mean much to me. What are you trying to say? I am not sure. To me, a performance result would at least mean that you are comparing some statistics (e.g. valid minutes per day, valid days per person, non-collection rates, etc.) I did not see anything here.

Also, in the activity detection algorithm, did the authors think to compare turning geolocation tracking on and off to the data with subject self-reports (i.e., the gold standard) to assess how well it performs? In my opinion, this would be a more important analysis to find out how well the activity recognition works, rather than simply creating a bar chart with the distributions of the length of distances.

Overall, I do not see any comparison between the performance of these apps and other apps developed for these purposes. For example, Onnela et al. at Harvard developed an app called Beiwe that is very similar:

https://www.hsph.harvard.edu/onnela-lab/beiwe-research-platform/

I would recommend that the authors do more thorough research on what is already out there to discuss the pros and cons of this app in more detail.

Response 3: Thank you for bringing our attention to the Beiwe app and other similar apps. Our app is different in that it serves as a novel theoretical and methodological framework for collecting geolocation data of research participants, with a specific focus on maintaining privacy. However, we agree with the reviewer that readers may benefit from more context around other apps, and we have therefore listed some of the other apps like Beiwe when highlighting the difference of our app. (Page 2, Paragraph 3)

“Previous apps, such as Beiwe [16], bear similarities to this contribution, but did not focus solely on collecting geolocation data or evaluated the quality of the data collected. Another similar app called CareConekta [17] focused on the feasibility of using an app to collect GPS location. This app only collected 2 GPS heartbeats per day and the location accuracy was approximately 1km; however, our app continuously collects precise GPS data while the user is moving. Other studies have attempted to evaluate the accuracy of geolocation data collected from smartphones, but they did not account for energy consumption or private data handling.”

Since GPS receivers are considered standard practice, we did not perform additional testing of the locations tracked compared to self-reporting. We aim to build on proven components that can help streamline development and allow our findings to be situated within the context of previous work.

Regarding the comment about performance results, we have also included the valid tracking minutes per day in our results section (Page 11, Paragraph 5)

We have also included more explanation of the indoor/outdoor comparison:

“As seen in table 1, outdoor accuracy was on average 8.29 meters, which is slightly higher than the maximum GPS accuracy (5 meters). The difference can be due to signal blockage by buildings or trees and atmospheric conditions. Indoor accuracy had an average of 30.6 meters, with the 25th percentile at 3.1 meters, the 50th percentile at 14.2 meters, and the 75th percentile at 35 meters. The difference in accuracy can be largely attributed to the availability of WIFI and cellular service, impacting the precision of the location data.” (Page 12, Paragraph 1)

Regarding the comment about the activity recognition, our analysis aims to measure the distance traveled by the phone owner before the tracking is reactivated:

“As show in Figure 6, the distances traveled by the user before the tracking has resumed were under 200 meters, which is consistent with the stationary geofence of iOS as described in the methods section.” (Page 12, Paragraph 2)

Moreover, we noted in the limitations section that comparison with user reports data would strengthen this work:

“Future applications of our methodology must include a detailed analysis of data quality, specific to the group's characteristics and the cohort in question. For instance, if applied to a cohort with depression and limited mobility, the analysis may focus on indoor geolocation tracking accuracy, consider variations in smartphone technology, and tailor additional analyses to factors such as different socioeconomic statuses or physical activity levels. In addition, the data presented in the results section was collected from an iOS device, which might not be representative of the diverse range of operating systems and hardware configurations. Future research will be required to explore the discrepancies and effects across various operating systems and devices, as the current test may have neglected differences that could influence the functionality and results. Comparing the data collected by the app to the subjects’ self-reports would also strengthen this work.” (Page 15, Paragraph 3)

Reviewer 2 Report (New Reviewer)

This document presents an innovative mobile application for passive geolocation data collection in the field of mobile health (mHealth). The app enables efficient monitoring of users' location without draining their mobile device's battery. It utilizes Mobile Ads ID (MAID) to preserve users' privacy while sharing their geolocation data in health research. The integration of blockchain technology offers a transparent rewards system through smart contracts to incentivize active participation in research studies. The software tool facilitates data analysis and visualization of geolocation data for public health research. The results evaluate the accuracy of geolocation data and the performance of the activity recognition algorithm. Potential use cases in health are discussed, such as trajectory analysis to predict health outcomes and combining MAID with other sources of information. Although there are limitations, such as iOS restrictions and the need to improve privacy measures, the app represents a significant advancement in geolocation data collection in mHealth. The results of studies conducted with this app can have a meaningful impact on public health decision-making and the understanding of behavior-driven disease transmission. The work exhibits good writing fluency, and its results are a promising contribution. However, the document can be improved if the following recommendations are addressed:

The summary provides a concise summary of the purpose and key features of the application. However, you could benefit from providing more specific details on the research methodology and main findings. For example, mention the innovative aspects of the app, such as its energy efficiency tracking feature and the integration of blockchain technology to earn rewards. Add a short statement about the length of the study and the number of participants to give readers an idea of the scale of the research. Mention the main results, such as the accuracy of the geolocation data and the effectiveness of the activity recognition algorithm.

The introduction provides a clear overview of the purpose of the app and the potential benefits. However, it could benefit from a more engaging and compelling opening to grab the reader's attention early on. For example, consider adding a brief real-world scenario or statistics related to the importance of geolocation data in healthcare research to create a stronger hook in your introduction. Provide a clearer statement of the research objectives and research questions to set specific expectations for the reader.

The Materials and Methods Section provides extremely detailed technical information on application design and implementation, which can be overwhelming for non-technical readers. Suggestions, you can add a subsection or summary to the beginning of this section that describes the key design features and methodologies in simple terms, making it more accessible to a wider audience. Include a flowchart or diagram explaining the application's data collection and processing steps for ease of understanding.

The results section consists mainly of figures and a table but lacks a narrative linking them. Furthermore, the explanation of these figures and the table is very brief. Provide a more detailed explanation of the figures and table in the text, explaining their implications for the investigation and how they relate to application performance.

The Discussion Section does a good job of highlighting the newness of the app and potential use cases. However, it would benefit from a more critical analysis of the limitations and implications of the findings. It is suggested that the limitations section be expanded to address potential sources of bias or confounding in the data collection and analysis process. Discuss the broader implications of the app's design and findings for the field of mHealth, including potential ethical considerations and future research directions.

The Conclusion section provides a concise summary of the benefits and potential impact of the application. However, it lacks a compelling call to action or closing statement. Suggestions: Consider ending the conclusion with a strong statement that emphasizes the importance of geolocation data in advancing healthcare research and encouraging further adoption and development of similar apps in the field of mHealth.

Author Response

Kindly view the attached file to view the responses in the intended format.

Response to Reviewer 2 Comments

Point 1: The summary provides a concise summary of the purpose and key features of the application. However, you could benefit from providing more specific details on the research methodology and main findings. For example, mention the innovative aspects of the app, such as its energy efficiency tracking feature and the integration of blockchain technology to earn rewards. Add a short statement about the length of the study and the number of participants to give readers an idea of the scale of the research. Mention the main results, such as the accuracy of the geolocation data and the effectiveness of the activity recognition algorithm.

Response 1: We thank you for your comment and have added more specific details on the research methodology and main findings to the abstract (Page 1, Paragraph 1)

“In addition, our app leverages blockchain and traditional payments to streamline the compensations and has an intuitive user interface to encourage participation in research. The mobile tracking app was tested for 20 days on an iPhone 14 Pro Max, finding it accurately captured location during movement and promptly resumed tracking after inactivity periods, while consuming low percentage of battery life while running in the background.”

Point 2: The introduction provides a clear overview of the purpose of the app and the potential benefits. However, it could benefit from a more engaging and compelling opening to grab the reader's attention early on. For example, consider adding a brief real-world scenario or statistics related to the importance of geolocation data in healthcare research to create a stronger hook in your introduction. Provide a clearer statement of the research objectives and research questions to set specific expectations for the reader.

Response 2: We thank you for your suggestion and have attempted to grab the reader's attention using a real-world scenario to the introduction (Page 2, Paragraph 2)

“Soon after the start of the COVID-19 pandemic, it become evident that geolocation data would play an increasing role in public health, statistics, and disease outbreaks. Companies such as SafeGraph began partnering with academic institutions, and a large number of high impact manuscripts started being published showing the contributions of geolocation data on public health [11]. These studies collectively underline the potential of geolocation tracking data in linking environmental exposures such as air pollutants, NDVI, and walkability to health outcomes. However, they also highlight the need for addressing challenges related to technical problems, statistical methodologies, participant privacy, and security.”

Point 3: The Materials and Methods Section provides extremely detailed technical information on application design and implementation, which can be overwhelming for non-technical readers. Suggestions, you can add a subsection or summary to the beginning of this section that describes the key design features and methodologies in simple terms, making it more accessible to a wider audience. Include a flowchart or diagram explaining the application's data collection and processing steps for ease of understanding.

Response 3: We thank you for your suggestion and have both added a summery paragraph at the beginning of the methods section (Page 3, Paragraph 1) and added a flowchart.

“This section outlines the development of an app designed to record users' locations, focusing on both accuracy and energy efficiency. A novel algorithm is implemented to switch the GPS receiver's power on and off based on the phone's motion status, ensuring continuous tracking while conserving battery life. In addition, we describe a cloud-based database architecture to securely store large volumes of data and allow authorized entities to access it for analysis. Finally, we present a description of a built-in rewards system that utilizes a payment Application Programming Interface (API) and Cryptocurrencies.”

Point 4: The results section consists mainly of figures and a table but lacks a narrative linking them. Furthermore, the explanation of these figures and the table is very brief. Provide a more detailed explanation of the figures and table in the text, explaining their implications for the investigation and how they relate to application performance.

Response 4: Thank you for your feedback, we have attempted to address the reviewer’s feedback by providing more descriptions of the figures and explaining their implications in the results.

“As seen in Table 1, outdoor accuracy was on average 8.29 meters, which is slightly higher than the maximum GPS accuracy (5 meters). The difference can be due to signal blockage by buildings or trees and atmospheric conditions. Indoor accuracy had an average of 30.6 meters, with the 25th percentile at 3.1 meters, the 50th percentile at 14.2 meters, and the 75th percentile at 35 meters. The difference in accuracy can be largely attributed to the variability in WIFI and cellular service strength, impacting the precision of the location data.” (Page 12, Paragraph 1)

“As show in Figure 6, when our tracking stops and restarts again, the user has moved less than 200 meters. That confirms the tracking restarts within the stationary geofence of iOS as described in the methods section.” (Page 12, Paragraph 2)

“As shown in Figure 7, the number of hours reported by the iPhone Settings App was consistent with the number of valid tracking minutes reported in Figure 4.” (Page 13, Paragraph 1)

Point 5: The Discussion Section does a good job of highlighting the newness of the app and potential use cases. However, it would benefit from a more critical analysis of the limitations and implications of the findings. It is suggested that the limitations section be expanded to address potential sources of bias or confounding in the data collection and analysis process. Discuss the broader implications of the app's design and findings for the field of mHealth, including potential ethical considerations and future research directions.

Response 5: As suggested, we have attempted to address our limitations by expanding the discussion section to uncover analysis methods (Page 14, Paragraph 2) and ethical implications (Page 14, Paragraph 5) for using geolocation data in public health research.

“In the context of research, public health researchers, who utilize this app to gather research data, are responsible to inform subjects of potential effects and ethical considerations especially for individual users. This app, like the ones used for contact tracing during the COVID 19 pandemic or personalized exposure assessment, provides opportunities to collect real time data on movement, health status and environmental exposures [62, 63]. Certain features have been implemented in the app to ensure the users are informed about potential risks of using the app, such as the studies consent form and ability to stop tracking temporarily. However, the decision to implement apps for data collection needs consideration of the potential risks and benefits for individual users. The impact on individuals may involve increased surveillance and control measures, potential privacy risks and the chance of misinterpreting data which could lead to unintended harms to data owners [64]. Ethical considerations should also be a priority in terms of transparency in app usage plans decision making processes regarding implementation informing individuals about risks or benefits they may face well as measures to ensure confidentiality and consent when handling collected data [65]. Given the nature of data collection through apps it is crucial to establish a strong ethical framework that respects individual autonomy while maximizing technology’s potential benefits.” (Page 15, Paragraph 1)

In addition, we have further explained the (Page 15, Paragraph 3)

“Future applications of our methodology must include a detailed analysis of data quality, specific to the group's characteristics and the cohort in question. For instance, if applied to a cohort with depression and limited mobility, the analysis may focus on indoor geolocation tracking accuracy, consider variations in smartphone technology, and tailor additional analyses to factors such as different socioeconomic statuses or physical activity levels. In addition, the data presented in the results section was collected from an iOS device, which might not be representative of the diverse range of operating systems and hardware configurations. Future research will be required to explore the discrepancies and effects across various operating systems and devices, as the current test may have neglected differences that could influence the functionality and results. Comparing the data collected by the app to the subjects’ self-reports would also strengthen this work.”

Point 6: The Conclusion section provides a concise summary of the benefits and potential impact of the application. However, it lacks a compelling call to action or closing statement. Suggestions: Consider ending the conclusion with a strong statement that emphasizes the importance of geolocation data in advancing healthcare research and encouraging further adoption and development of similar apps in the field of mHealth.

Response 6: We thank you for your suggestion and have attempted to add a compelling call to action incorporating the reviewer’s suggestion (Page 16, Paragraph 1)

“Although still in its infancy, research using geolocation data holds promise in being able to rapidly and profoundly improve surveillance and delivery of public health and medicine.”

Reviewer 3 Report (New Reviewer)

Interesting contribution .

A discussion of the potential effects of the use of the app for the individuals must be done. The focus of the manuscript is on the possibility to collect research data.  The single individuals, the effects, consequences and the ethical impact must be analyzed and considered.  

How the app will be used. who will make the decision to implement the app as a tool to collect data , how to inform individuals of eventual risks or benefits to register data are not mentioned in the manuscript. 

Author Response

Response to Reviewer 3 Comments

Point 1: Interesting contribution .

A discussion of the potential effects of the use of the app for the individuals must be done. The focus of the manuscript is on the possibility to collect research data.  The single individuals, the effects, consequences and the ethical impact must be analyzed and considered.  

How the app will be used. who will make the decision to implement the app as a tool to collect data , how to inform individuals of eventual risks or benefits to register data are not mentioned in the manuscript. 

Response 1: We thank you for your feedback and have added a subsection of the Discussion section to discuss the ethical considerations highlighted by the reviewer for using mobility data in public health research (Page 15, Paragraph 2):

"In the context of research, public health researchers, who utilize this app to gather research data, are responsible to inform subjects of potential effects and ethical considerations especially for individual users. This app, like the ones used for contact tracing during the COVID 19 pandemic or personalized exposure assessment, provides opportunities to collect real time data on movement, health status and environmental exposures [62, 63]. Certain features have been implemented in the app to ensure the users are informed about potential risks of using the app, such as the studies consent form and ability to stop tracking temporarily. However, the decision to implement apps for data collection needs consideration of the potential risks and benefits for individual users. The impact on individuals may involve increased surveillance and control measures, potential privacy risks and the chance of misinterpreting data which could lead to unintended harms to data owners [64]. Ethical considerations should also be a priority in terms of transparency in app usage plans, decision-making processes regarding implementation, informing individuals about risks or benefits they may face, as well as measures to ensure confidentiality and consent when handling collected data [65]. Given the nature of data collection through apps it is crucial to establish a strong ethical framework that respects individual autonomy while maximizing technology’s potential benefits.”

Round 2

Reviewer 1 Report (New Reviewer)

The authors have successfully addressed all my concerns. I have no more comments.

This manuscript is a resubmission of an earlier submission. The following is a list of the peer review reports and author responses from that submission.

Round 1

Reviewer 1 Report

The review results of the article are shared below. Considering these issues, it is thought that the corrections to be made will make the contrubition more mature and satisfactory.

1-) It is seen that the developed App within the scope of the study will use many sensor data. It should be examined whether these will constitute an ethical violation in terms of collecting and sharing personal data.

2-) There are many applications that do what is intended in the study. What kind of innovation the algorithm (application) discussed here will bring to the existing ones or what gap it will fill must be revealed.

It should be revealed why this application should be used rather than one of the dozens of existing applications.

3-) Accuracy analysis of the study must be conducted with a realistic field application (both indoor and outdoor enviroments).

Author Response

Point 1: It is seen that the developed App within the scope of the study will use many sensor data. It should be examined whether these will constitute an ethical violation in terms of collecting and sharing personal data.

Response 1: As suggested, we have addressed the ethical concerns related to collecting geolocation data in the discussion section. The App deploys data encryption and anonymized data schema to ensure the collected data is not leaked. Moreover, we provide users with consent forms and data visualization features to explain the data being collected and how it is used. We also conducted a separate study that interviewed opioid users and incorporated their feedback concerning privacy.

Point 2: There are many applications that do what is intended in the study. What kind of innovation the algorithm (application) discussed here will bring to the existing ones or what gap it will fill must be revealed.

It should be revealed why this application should be used rather than one of the dozens of existing applications.

Response 2: Thank you for your constructive comment. While it is true that numerous applications exist with similar functionalities, our contribution distinguishes itself through the unique amalgamation of battery-efficient tracking logic, a scalable cloud architecture, and the use of Mobile Ads ID (MAID) for data privacy. Most of the existing applications tend to consume substantial battery power due to continuous tracking, but our algorithm optimizes the use of smartphone sensors to efficiently monitor physical activity and collect geolocation data, significantly improving battery performance. Additionally, our scalable cloud architecture not only allows for effective handling and storage of geolocation data but also proficiently manages cost, thus offering a solution that is economically viable for long-term and large-scale deployment. Lastly, our use of MAID represents a significant leap in preserving data privacy, integrity, and usability in healthcare research. It offers a de-identification method that respects user privacy while still providing usable data for research. In essence, the combination of these three aspects—battery efficiency, cost-effective scalability, and data privacy—creates an application that fills a critical gap in the existing technology landscape.

Moreover, we are not aware of any other apps that collectes this type of data for research. Most of the other apps are developed by corparations, which makes it harder to get access to the data or code base.

Point 3: Accuracy analysis of the study must be conducted with a realistic field application (both indoor and outdoor enviroments).

Response 3: We thank you for your comment and provided the accuracy ranges of the indoor and outdoor GPS receiver in Table 1. In a separate study, we also conducted interviews with actual users of the app and incorporated their feedback into the app.

Reviewer 2 Report

Based on the review of this paper, we find it necessary to reject the manuscript due to several shortcomings. The lack of substantial results and the overall quality of the paper diminish its scientific relevance and fail to contribute to the advancement of scientific knowledge.

Despite these limitations, it is important to note that the topic addressed in the manuscript holds significant interest. With further improvements and revisions, the author should consider resubmitting the manuscript. Doing so would provide an opportunity to enhance the scientific value and strengthen the impact of the research.

Below a comprehensive list of major and minor points to be taken into account for the resubmission of manuscript:

Major Points:

  1. The author needs to include a dedicated results section in this paper.

  2. Although the paper presents an interesting application that could be helpful for researchers, the lack of access to the code and backend infrastructure limits its potential usability for other researchers.

  3. While the idea of implementing a reward system using Blockchain is innovative, it appears to be disconnected from the main research focus.

  4. The paper generally still has room for in depth explanations about the sensor usage. Sensor frequencies, the app’s precise energy consumption in comparison to an idle running phone, and actual test results or a demonstration of the app itself are missing.

  5. The conclusion is too generic and does not effectively summarize the research findings. Moreover, what are the research findings?

Minor Points:

  1. Line 45: Small mistake in repeating “geolocation data” twice

  2. Line 46: Please clarify what is meant by "Alcohol Points of Interest."

  3. Line 76: It would be highly recommended to include an energy consumption comparison (e.g. idle vs. running) in the paper.

  4. Line 103: Please define the criteria for "Low" and "High" energy.

  5. Line 128: Provide more details about the framework's implementation, including the methods used and how it operates. Additionally, specify the sampling rate for GPS data retrieval, both in the worst case and best case scenarios (e.g., frequency in seconds).

  6. Line 157: Is the backend (API) code publicly available? Consider making it more reusable for others, for example, by providing a Docker container.

  7. Line 177: Does the author believe that using only HTTPS encryption is sufficient for protecting health information? Why is additional AES encryption not implemented? Is this a potential limitation?

  8. Line 248: The terms “should”, “intuitive”, and “easy” are supposed to clarify that indexing latitude and longitude is the best way to store geolocation data. Yet, an explanation, or a reference, clarifying these statements is missing. Thereby, the reader is forced to trust the authors in their assessment.

  9. Line 293: The use case diagram contains elements that are not used in use case diagrams, rendering it unclear and not useful for readers. In its current state, it contains elements of activity diagrams, use-case diagrams, and flow charts but combines them using unknown conventions and rules. It is recommended to decide on one type of diagram and stick to the UML conventions.

  10. Line 294: Figure 3 only shows a dashboard and settings view, but lacks detailed information about the data and how the reward system functions.

  11. Line 300: It would be helpful to include screenshots for the four pages mentioned.

  12. Line 310: While payment via Cryptocurrencies or NFTs is an innovative approach, it lacks practicality for the use case of this app. It would require all participants to 1) own a crypto wallet (hard- or soft) to redeem their tokens, 2) understand the concept of NFTs and minting them, 3) provide equal value to the participants as the fiat currency issued by their state of origin. Unless the app is rolled out to a crypto endorsing test group, each of the priorly mentioned points speaks strongly against crypto payments of any kind in this use case and for a classical payment approach.
    Even though some of these points are mentioned in line 323, the authors could be more precise about the actual drawbacks.

  13. Line 347: The Discussion section should address the limitations and potential future work of the research.

  14. Line 381: The benefit of Figure 4 is not clear. Please provide a clearer explanation or consider removing it.

  15. Line 389: Why aren't GPS / Geolocation privacy measures implemented in the app? Please discuss this aspect.

  16. Line 435: The screenshots showing the onboarding process do not contribute to a better understanding of the application's functionality. Consider removing them or replacing them with more relevant visuals.

see below

Author Response

Point 1: The author needs to include a dedicated results section in this paper

Response 1: While the call for a dedicated results section is a standard requirement in research papers, it's important to recognize that the nature of the paper in question differs from typical empirical research. This paper presents a novel method for tracking subjects for research while maintaining privacy, which is largely theoretical and methodological in its approach. Its primary objective is to propose and articulate this new approach, rather than to present empirical results of its application. In such cases, the emphasis is often on the theoretical framework, practicality, potential applications, and possible implications for future research, rather than on immediate empirical results. Consequently, expecting a conventional results section might not align with the context and objectives of this paper. Nonetheless, any subsequent paper that applies this methodology in a practical scenario should indeed include a dedicated results section.

Point 2: Although the paper presents an interesting application that could be helpful for researchers, the lack of access to the code and backend infrastructure limits its potential usability for other researchers.

Response 2: We thank you for your suggestion and added a statement for sharing code: code is available upon request from the authors.

Point 3: While the idea of implementing a reward system using Blockchain is innovative, it appears to be disconnected from the main research focus.

Response 3: We thank you for your comment. While the broader focus of the paper is on collecting geolocation data, using blockchain as a form of collection is an essential part of this process since it makes it more likely that the study participants will be more likely to provide the access needed to collect this data. We think it is important for the replication of this research that others be aware of this form of compensation.

Point 4: The paper generally still has room for in depth explanations about the sensor usage. Sensor frequencies, the app’s precise energy consumption in comparison to an idle running phone, and actual test results or a demonstration of the app itself are missing.

Response 4: As suggested, we included details about the energy consumption in ideal state and sampling rate.

Point 5: The conclusion is too generic and does not effectively summarize the research findings. Moreover, what are the research findings?

Response 5: As suggested, we have summarized the research findings in the conclusion.

Point 6: Line 45: Small mistake in repeating “geolocation data” twice

Response 6: We thank you for your suggestion and have removed the typo.

Point 7: Line 46: Please clarify what is meant by "Alcohol Points of Interest."

Response 7: As suggested, we renamed it to locations associated with alcohol consumption.

Point 8: Line 76: It would be highly recommended to include an energy consumption comparison (e.g. idle vs. running) in the paper.

Response 8: We thank you for your comment and addressed this point in line 145.

Point 9: Line 103: Please define the criteria for "Low" and "High" energy.

Response 9: As suggested, we provided definitions of low and high energy consumption in the paragraph above the table.

Point 10: Line 128: Provide more details about the framework's implementation, including the methods used and how it operates. Additionally, specify the sampling rate for GPS data retrieval, both in the worst case and best case scenarios (e.g., frequency in seconds).

Response 10: As suggested, we explained the framework implementation. Moreover, the sampling rate is based on distance rather than time. The minimum sampling rate is 5 meters.

Point 11: Line 157: Is the backend (API) code publicly available? Consider making it more reusable for others, for example, by providing a Docker container.

Response 11: We thank you for your suggestion and added a statement saying that code is available upon request from the authors.

Point 12: Line 177: Does the author believe that using only HTTPS encryption is sufficient for protecting health information? Why is additional AES encryption not implemented? Is this a potential limitation?

Response 12: We thank you for your suggestion and added this to the future work section. HTTPS, or HTTP over SSL/TLS, does provide a strong layer of security in the transmission of data over the Internet by encrypting the data in transit. It essentially ensures that the connection between the user's device and the server is secure. The data server will only be accessible by the research team; thus, while desired as future work, AES can be substituted by HTTPS.

Point 13: Line 248: The terms “should”, “intuitive”, and “easy” are supposed to clarify that indexing latitude and longitude is the best way to store geolocation data. Yet, an explanation, or a reference, clarifying these statements is missing. Thereby, the reader is forced to trust the authors in their assessment.

Response 13: As suggested, we added more references and explanations to this paragraph.

Point 14: Line 293: The use case diagram contains elements that are not used in use case diagrams, rendering it unclear and not useful for readers. In its current state, it contains elements of activity diagrams, use-case diagrams, and flow charts but combines them using unknown conventions and rules. It is recommended to decide on one type of diagram and stick to the UML conventions.

Response 14: We thank you for your suggestions and have changed the use case diagram into an activity diagram to follow the UML conventions.

Point 15: Line 294: Figure 3 only shows a dashboard and settings view, but lacks detailed information about the data and how the reward system functions.

Response 15: As suggested, we have added screenshot of the data visualization in the appendix. Moreover, we provide explanation of how the payment system works on page 6.

Point 16: Line 300: It would be helpful to include screenshots for the four pages mentioned.

Response 16: As suggested, we included screenshot of the data visualization in the appendix.

Point 17: Line 310: While payment via Cryptocurrencies or NFTs is an innovative approach, it lacks practicality for the use case of this app. It would require all participants to 1) own a crypto wallet (hard- or soft) to redeem their tokens, 2) understand the concept of NFTs and minting them, 3) provide equal value to the participants as the fiat currency issued by their state of origin. Unless the app is rolled out to a crypto endorsing test group, each of the priorly mentioned points speaks strongly against crypto payments of any kind in this use case and for a classical payment approach.
Even though some of these points are mentioned in line 323, the authors could be more precise about the actual drawbacks.

Response 17: As suggested, we recognize that only specific group of people would be interested in blockchain as a compensation method; thus we are offering alternative payment method.

Point 18: Line 347: The Discussion section should address the limitations and potential future work of the research.

Response 18: As suggested, we added limitations and future work to the discussion.

Point 19: Line 381: The benefit of Figure 4 is not clear. Please provide a clearer explanation or consider removing it.

Response 19: As suggested, we have provided a clearer explanation of the purpose of the figure.

Point 20: Line 389: Why aren't GPS / Geolocation privacy measures implemented in the app? Please discuss this aspect.

Response 20: We thank you for your comment and have addressed the ethical concerns related to collecting geolocation data in the discussion section. We also conducted a separate study about the ethical concerns of using this app. Even sensitive populations were willing to participate in this type of data collection. We also incorporated the feedback we got from the recruited participants (opioid users).

Point 21: Line 435: The screenshots showing the onboarding process do not contribute to a better understanding of the application's functionality. Consider removing them or replacing them with more relevant visuals.

Response 21: As suggested, we replaced the signup page screenshot with the data visualization screenshot.

Round 2

Reviewer 1 Report

The answers to the suggestions I have made have been satisfactory. It is appropriate to publish it as it is.

Reviewer 2 Report

The scientific merit of this paper is not present as no data or results are published. Furthermore, the approach presented in this paper lacks novelty and fails to contribute to the existing body of scientific knowledge. Consequently, it does not meet the standards required for a publication in a peer-reviewed journal and cannot be considered a scientific paper.

-